# Improving Auto-Augment via Augmentation-Wise Weight Sharing

**Keyu Tian**
SenseTime Research
Beihang University
tiankeyu.00@gmail.com

**Chen Lin**
SenseTime Research
linchen@sensetime.com

**Ming Sun**
SenseTime Research
sunming1@sensetime.com

**Luping Zhou**
University of Sydney
luping.zhou@sydney.edu.au

**Junjie Yan**
SenseTime Research
yanjunjie@sensetime.com

**Wanli Ouyang**
University of Sydney
wanli.ouyang@sydney.edu.au

## Abstract

The recent progress on automatically searching augmentation policies has boosted the performance substantially for various tasks. A key component of automatic augmentation search is the evaluation process for a particular augmentation policy, which is utilized to return reward and usually runs thousands of times. A plain evaluation process, which includes full model training and validation, would be time-consuming. To achieve efficiency, many choose to sacrifice evaluation reliability for speed. In this paper, we dive into the dynamics of augmented training of the model. This inspires us to design a powerful and efficient proxy task based on the **Augmentation-Wise Weight Sharing (AWS)** to form a fast yet accurate evaluation process in an elegant way. Comprehensive analysis verifies the superiority of this approach in terms of effectiveness and efficiency. The augmentation policies found by our method achieve superior accuracies compared with existing auto-augmentation search methods. On CIFAR-10, we achieve a top-1 error rate of 1.24%, which is currently the best performing single model without extra training data. On ImageNet, we get a top-1 error rate of 20.36% for ResNet-50, which leads to 3.34% absolute error rate reduction over the baseline augmentation.

## 1   Introduction

Deep learning techniques have been heavily utilized in the computer vision area and made remarkable progress in lots of tasks, such as image classification [16, 34, 40], object detection [23, 28, 18, 24], segmentation [2, 11], image captioning [36], and human pose estimation [35]. Overfit is a commonly acknowledged issue of deep learning algorithms. Various Regularization techniques are proposed in different tasks to fight overfit. Data augmentation, which increases both the amount and the diversity of the data by applying semantic invariant image transformations to training samples [34, 1], is the most commonly used regularization due to its simplicity and effectiveness. There are various frequently used augmentation operations for image data, including traditional image transformations such as resizing, cropping, shearing, horizontal flipping, translation, and rotation. Recently, several special operations, such as Cutout [7] and Sample Pairing [14], are also proposed. It has been widely

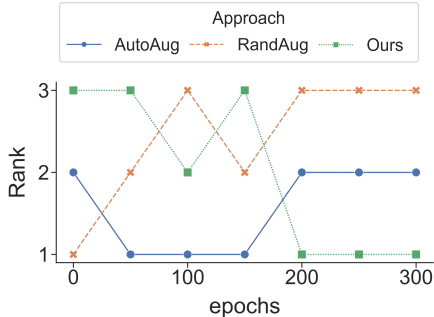

Figure 1: **An investigation of the change of rankings in augmented training.** We train ResNet-18 [12] on CIFAR-10 [15] for 300 epochs in total, utilizing different augmentation strategies ([4], [5], and ours).

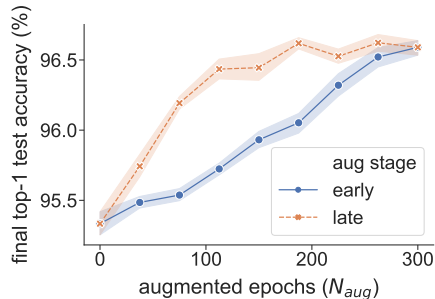

Figure 2: **An investigation of the relationship between the performance gains and the augmented training periods**. We apply augmentation [4] in the start or the end $N_{aug}$ epochs.

observed [17, 16, 37] that augmentation strategies influence the final performances of deep learning models considerably.

However, choosing appropriate data augmentation strategies is time-consuming and requires extensive efforts from experienced human experts. Hence automatic augmentation techniques [4, 5, 20, 13, 21, 41] are leveraged to search for performant augmentation strategy according to specific datasets and models. Numerous experiments show that these searched policies are superior to hand-crafted policies in many computer vision tasks. These techniques design different evaluation processes to conduct searches.

The most straightforward approach [4] use a plain evaluation process which fully trains the model with different augmentation policies repeatedly to obtain the reward for reinforcement learning agent. Inevitably, this approach raises the time-consuming issue as it requires a tremendous amount of computational resources to train thousands of child models to complete.

To alleviate the computational cost, most of the efficient works [13, 21, 41] utilize the joint optimization approach to evaluate the strategies every few iterations, getting rid of training multiple networks from scratch repeatedly. Although being efficient, most of these methods only have the performance similar to that of [5] due to the compromised evaluation process. Specifically, the compromised evaluation process would distort the ranking for augmentation strategies since the ranks of the models trained with too few iterations are known to be inconsistent with the final modelss trained with sufficient iterations. Fig. 1 shows this phenomenon, where the relative ranks change a lot during the whole training process.

An ideal evaluation process should be efficient as well as highly reliable to produce accurate rewards for augmentation strategies. In order to achieve this, we dive into the training dynamics with different data augmentations. We observe that the augmentation operations in the later training period are more influential. Based on this, we design a new evaluation process, which is a proxy task with an Augmentation-wise Weight Sharing (AWS) strategy. Compared with [4], we improve efficiency significantly via this weight sharing strategy and make it affordable to directly search on large scale datasets. And the performance gains are also substantial. Compared with previous efficient methods, our method produces more reliable evaluation shown in Sec. 4.4 with competitive computation resources. Our main contribution can be summarized as follows: 1) We propose an efficient yet reliable proxy task utilizing a novel augmentation-wise weight sharing strategy to be the evaluation process for augmentation search methods. 2) We design a new search pipeline for auto-augmentation search utilizing the proposed proxy task and achieved superior accuracy compared with existing auto-augmentation search methods.

The augmentation policies found by our approach achieve outstanding performance. On CIFAR-10, we achieve a top-1 error rate of 1.24%, which is the currently best-performing single model without extra training data. On ImageNet, we get a top-1 error rate of 20.36% for ResNet-50, which leads to 3.34% improvement over the baseline augmentation. The augmentation policies we found on both CIFAR and ImageNet benchmark will be released to the public as an off-the-shelf augmentation policy to push the boundary of the state-of-the-art performance.

## 2  Related Work

### 2.1  Auto Machine Learning and Neural Architecture Search

Auto Machine Learning (AutoML) aims to free human practitioners and researchers from these menial tasks. Recent advances focus on automatically searching neural network architectures. One of the first attempts [43, 42] was utilizing reinforcement learning to train a controller representing a policy to generate a sequence of symbols representing the network architecture. An alternative to reinforcement learning is evolutionary algorithms, that evolved the topology of architectures by mutating the best architectures found so far [27, 38, 30, 19, 9]. Recent efforts such as [22, 25, 26], utilized several techniques trying to reduce the search cost. Note that [4] utilized a similar controller inspired by [43], whose training is time-consuming, to guide the augmentation policy search. Our auto-augmentation search strategy is much more efficient and economical compared to it.

### 2.2  Automatic Augmentation

Recently, some automatic augmentation approaches [4, 20, 13, 21, 5, 41] have been proposed. The common purpose of them is to search for powerful augmentation policies automatically, by which the performances of deep models can be enhanced further. [4] formulates the automatic augmentation policy search as a discrete search problem and employs a reinforcement learning framework to search the policy consisting of possible augmentation operations, which is most closely related to our work. Our proposed method also optimizes the distribution of the discrete augmentation operations, but it is much more computationally economical, benefiting from the weight sharing technique. Much other previous work [13, 21, 41] takes the single-step approximation to reduce the computational cost dramatically by getting rid of training multiple networks.

## 3  Method

### 3.1  Motivations

#### 3.1.1  Key Observation

As a powerful regularization technique, data augmentation is applied to relief overfitting [32]. Another popular regularization method is early stopping [29], meaning to compute the validation error periodically, and stop training when the validation error starts to go up. It shows the overfitting phenomenon may mostly occur in the late stages of the training. Thus, a natural conjecture could be raised: data augmentation improves the generalization of the model, mainly in the *later* training process.

To investigate and verify this, we explore the relationship between the performance gains and the augmented periods. We train ResNet-18 [12] on CIFAR-10 [15] for 300 epochs in total, some of which are augmented by the searched policy of AutoAug [4]. Specifically, we apply augmentation in the start or the end $N_{aug}$ epochs, where $N_{aug}$ denotes the number of epochs with augmentation. We repeat each experiment eight times to ensure reliability. The result is shown in Fig. 2, which indicates two main pieces of evidence as follows: 1) With the same number of the augmented epochs $N_{aug}$, applying data augmentation in the *later stages* can constantly get better model performance, as the dashed curve is always above the solid one. 2) In order to train models to the same level of performance, conducting data augmentation in the *later stages* requires fewer epochs of augmentation compared with conducting it in the early stages, as the dashed curve is always on the left of the solid one.

In sum, our empirical results show that data augmentation functions more in the late training stages, which could be took advantage of to produce efficient and reliable reward estimation for different augmentation strategies.

#### 3.1.2  Augmentation-Wise Weight Sharing

Inspired by our observation, we propose a new proxy task for automatic augmentation. It consists of two stages. In the first stage, we choose a shared augmentation strategy to train the shared weights, namely, the *augmentation-wise* shared model weights. We borrow the weight sharing paradigm from

NAS that shares weights among different network architectures to speed up the search. Please note that, to the best of our knowledge, this is the first work to investigate the weight sharing technique for automatic augmentation search. In the second stage, we conduct the policy search efficiently. Reliability remains as augmentation operations function more in the late stages. And experiments in Sec. 4.4 also verify this.

## 3.2 Auto-Aug Formulation

Auto-Aug aims to find a set of augmentation operations for training data, which maximize the performance of a deep model. In this work, we denote training set as $\mathcal{D}_{tr}$ , validation set as $\mathcal{D}_{val}$. We use $x$ and $y$ to denote the image and its label. Here we are searching data augmentation strategy for a specific model denoted as $\mathcal{M}_\omega$, which is parameterized by $\omega$. We regard our augmentation strategy as a distribution $p_\theta(O)$ over candidate image transformations, which is controlled by $\theta$. $O$ denotes the set of operations. More detailed construction on the augmentation policy space would be introduced in Sec. 3.4.

The objective of obtaining the best augmentation policy (solving for $\theta$) could be described as a bilevel optimization problem. The inner level is the *model weight optimization*, which is solving for the optimal $\omega^*$ given a fixed augmentation policy $\theta$

$$\omega^* = \arg\min_\omega \frac{1}{\#\mathcal{D}_{tr}} \sum_{(x,y)\in\mathcal{D}_{tr}} \mathbb{E}_{O\sim p_\theta} \mathcal{L}(M_\omega(O(x)), y) , \tag{1}$$

where $\mathcal{L}$ denotes the loss function.

The outer level is the *augmentation policy optimization*, which is optimizing the policy parameter $\theta$ given the result of the inner level problem. Notably, the objective for the optimization of $\theta$ is the validation accuracy ACC

$$\theta^* = \arg\max_\theta \text{ACC}(\omega^*, \mathcal{D}_{val}) , \tag{2}$$

where $\theta^*$ denotes the parameter of the optimal policy and $\text{ACC}(\omega^*, \mathcal{D}_{val})$ denotes the validation accuracy obtained by $\omega^*$. This problem is a typical bilevel optimization problem [3]. Solving for the inner loop is extremely time-consuming. Thus, it is almost impossible to generalize this approach to a large scale dataset without compromise [4]. More recent works focusing on reducing the time complexity for solving bilevel optimization problems [13, 21, 41] have been proposed. They take a single-step approximation borrowing from NAS literature [22] to avoid training multiple networks from scratch. Instead of solving the inner level problem, single-step approximation takes only one step for $\omega_t$ based on previous $\omega_{t-1}$, and utilizes $\omega_t$, which approximates the solution of the inner level problem, to update $\theta$. These approaches are empirically efficient, but [5] shows it is possible to achieve compatible or even stronger performance using a random augmentation policy. Thus, a new approach to perform an efficient auto-augmentation search is desirable.

## 3.3 Our Proxy Task

Inspired by our observation that the later augmentation operations are more influential than the early ones, in this paper, we propose a new proxy task that substitutes the process of solving the inner level optimization by a computational efficient evaluation process.

The basic idea of our proxy task is to partition the augmented training of the network parameters $\omega$ (i.e., the inner level optimization) into two parts. In the first part (i.e., the early stage) a shared augmentation policy is applied to training the network regardless of the current policy $\theta$ given by the outer level optimization; and in the second part (i.e., the late stage), the network model is fine-tuned from the augmentation-wise shared weights by the given policy $\theta$ so that it could be used to evaluate the performance of this policy. Since the shared augmented training in the first part is independent of the given policy $\theta$, it only needs to be trained once for all candidate augmentation policies to search, which significantly speeds up the optimization. We call this strategy the *augmentation-wise weight sharing*.

Now our problem boils down to find a good shared augmentation policy $\bar{\theta}$ for the first part training. In the following, we show that this could be trivially obtained via the following proposition.

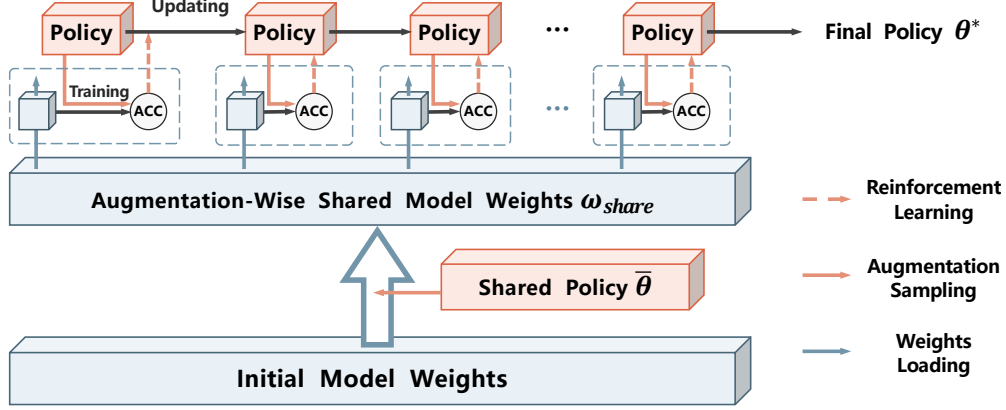

Figure 3: **The overview of our method.** Firstly, we train the model with the shared augmentation policy $\bar{\theta}$ to get the augmentation-wise shared weights $\omega_{share}$. Then we fine-tune it repeatedly and use ACC($\bar{\omega}_\theta^*$) to update the policy under the searching.

**Proposition.** Let $\tau = [\tau_1, \tau_2, \cdots, \tau_N]$ denote an arbitrary augmentation trajectory consisting of $N$ augmentation operations. Let $p_{\theta\theta}$ and $p_{\bar{\theta}\theta}$ be the trajectory distributions without or with the augmentation-wise weight sharing, that is: $p_{\theta\theta}(\tau) = \Pi_{i=1}^N p_\theta(\tau_i)$, and $p_{\bar{\theta}\theta}(\tau) = \Pi_{i=1}^K p_{\bar{\theta}}(\tau_i) \Pi_{i=K+1}^N p_\theta(\tau_i)$, where $K$ denotes the numbers of augmentation operations in the early training stage. Here $\bar{\theta}$ and $\theta$ indicate the shared and the given policy, respectively. The KL-divergence between $p_{\theta\theta}$ and $p_{\bar{\theta}\theta}$ is minimized when $\bar{\theta}$ is uniform sampling, i.e., $p_{\bar{\theta}}(\tau_1) = p_{\bar{\theta}}(\tau_2) = \cdots = p_{\bar{\theta}}(\tau_K)$ for all the possible $\tau_i$. The detailed proof is provided in the supplementary materials.

The above proposition tells us that with a simple uniform sampling for augmentation-wise weight sharing, the obtained augmentation trajectories would be similar to those without using the augmentation-wise weight sharing. This is a favorable property because of the follows. To enhance the reliability of the search algorithm, it is necessary to maintain a high correlation between ACC($\bar{\omega}_\theta^*$) and ACC($\omega^*$), where $\bar{\omega}_\theta^*$ indicates the network parameters trained by our proxy task. So, it is desired to make $p_{\text{alone}}(\omega) = \mathbb{E}_{\tau \sim p_{\theta\theta}}[p(\omega|\tau)]$ and $p_{\text{share}}(\omega) = \mathbb{E}_{\tau \sim p_{\bar{\theta}\theta}}[p(\omega|\tau)]$ as close as possible. We can achieve this by producing similar augmentation trajectories via employing a uniform sampling for the shared augmentation policy $\bar{\theta}$.

We use a uniform distribution $U(O)$ sampling the augmentation transforms to train the shared parameter checkpoint $\omega_{share}$:

$$\omega_{share} = \arg\min_\omega \frac{1}{\#\mathcal{D}_{tr}} \sum_{(x,y) \in \mathcal{D}_{tr}} \mathbb{E}_{O \sim U} \mathcal{L}(M_\omega(O(x)), y) . \tag{3}$$

In the second part training, to get the performance estimation for particular augmentation policy which has the parameter equals to $\theta$ we load $\omega_{share}$ and fine-tune the checkpoint with this augmentation policy. We denote the parameter obtained by finetuning with augmentation $\theta$ as $\bar{\omega}_\theta^*$. Note that the cost for obtaining $\bar{\omega}_\theta^*$ is very cheap compared with training from scratch. Thus, we optimize the augmentation policy parameters $\theta$ with

$$\theta^* = \arg\max_\theta \text{ACC}(\bar{\omega}_\theta^*, \mathcal{D}_{val}) . \tag{4}$$

In other words, we obtain $\omega_{share}$ once, then repeat $T_{max}$ times to reuse it and conduct the late training process for optimizing the policy parameter.

Moreover, by adjusting the number of epochs of finetuning, we can still maintain the reliability of policy evaluation to a large extent, which is verified in the supplementary material. In Sec. 4.4 we study the superiority of this proxy task, as we empirically find that there is a strong correlation between ACC($\bar{\omega}_\theta^*$) and ACC($\omega^*$).

---

**Algorithm 1** AWS Auto-Aug Search

---

**Inputs:** $\mathcal{D}_{tr}, \mathcal{D}_{val}, T_{max}$
Obtain $\omega_{share}$ in Equ. 3;
**while** $T \le T_{max}$ **do**
    Load $\omega_{share}$;
    Fine-tune $\omega_{share}$ to get $\bar{\omega}_\theta^*$;
    Use ACC$(\bar{\omega}_\theta^*, \mathcal{D}_{val})$ to update $\theta$;
**end while**
$\theta^* = \theta$;
return $\theta^*$;

---

### 3.4 Augmentation Policy Space and Search Pipeline

**Augmentation Policy Space**  In this paper, we regard the policy parameter $\theta$ as probability distributions on the possible augmentation operations. Let $K$ be the number of available data augmentation operations in the search space, and $O = \{\mathcal{O}^{(k)}(\cdot)\}_{k=1}^{K}$ be the set of candidates. Accordingly, each of them has a probability of being selected denoted by $p_\theta(\mathcal{O}^{(k)})$. For each training image, we sample an augmentation operation from the distribution of $\theta$, then apply to it. Each augmentation operation is a pair of augmentation elements. Following [4], we select the same augmentation elements, except Cutout [7] and Sample Pairing [14]. There are 36 different augmentation elements in total. The details are listed in the supplementary material.

More precisely, the augmentation distribution is a multinomial distribution which has $K$ possible outcomes. The probability of the $k$-th operation is a normalized sigmoid function of $\theta_k$. As a single augmentation operation is defined as a pair of two elements, resulting in $K = 36^2$ possible combinations (the same augmentation choice may repeat), we have $p_\theta(\mathcal{O}^{(k)}) = \frac{\frac{1}{1+e^{-\theta_k}}}{\sum_{i=1}^{K}(\frac{1}{1+e^{-\theta_i}})}$ .

**Search Pipeline**  As our proxy task is flexible, any heuristic search algorithm is applicable. In practical implementation, we empirically find that Proximal Policy Optimization [31] is good enough to find a good $\theta^*$ in Equ. 4. In practice, we also utilize the baseline trick [33] to reduce the variance of the gradient estimation. The baseline function is an exponential moving average of previous rewards with a weight of 0.9. The complete task pipeline using the augmentation-wise weight sharing technique is presented in Algorithm 1.

## 4 Experiments and Results

### 4.1 Datasets and Comparison Methods

Following the literature on automatic augmentation, we evaluate the performance of our proposed method on three classification datasets: CIFAR-10 [15], CIFAR-100 [15], and ImageNet [6]. The detailed description and splitting ways of these datasets are presented in the supplementary material. To fully demonstrate the advantage of our proposed method, we make a comprehensive comparison with the state-of-the-arts augmentation methods, includings Cutout [7], AutoAugment (AutoAug) [4], Fast AutoAugment (Fast AA) [20], OHL-Auto-Aug (OHL) [21], PBA [13], Rand Augment (RandAug) [5], and Adversarial AutoAugment (Adv. AA) [41].

### 4.2 Implementation Details

**CIFAR**  On CIFAR-10 and CIFAR-100, following the literature, we use ResNet-18 [12] and Wide-ResNet-28-10 [39], respectively, as the basic models to search the policies, and transfer the searched policies to other models, including to Shake-Shake (26 $2 \times 32$d) [8] and PyramidNet+ShakeDrop [10]. As mentioned, our training process is divided into two parts. The numbers of epochs of each part are set to 200 and 10, respectively, leading to 210 total number of epochs in the search process. The $T_{max}$ is set to 500. To optimize the policy, we use the Adam optimizer with a learning rate of $\eta_\theta = 0.1$, $\beta_1 = 0.5$ and $\beta_2 = 0.999$. Some other details are studied and reported in the supplementary.

**ImageNet**    During the policy search process, we use ResNet-50 [12] as the basic model, and then transfer the policies to ResNet-200 [12]. The learning rate $\eta_\theta$ is set to 0.2. The numbers of epochs of the two training stages are set to 150 and 5, respectively. Other hyper-parameters of the search process are the same as what we use for the CIFAR datasets. Some other details are studied and reported in the supplementary.

## 4.3    Comparison with the state-of-the-arts

The comparisons between our AWS method and the state-of-the-arts are reported in Tab. 1, Tab. 2 and Tab. 3. To minimize the influence of randomness, we run our method repetitively for eight times on CIFAR and four times on ImageNet, and report our test error rates in terms of Mean ± STD (standard deviation). For other methods in comparison, we directly quote their results from the original papers. Except Adv. AA [41], these methods only report the average test error rates. "Baseline" in tables refers to the basic models using only the default pre-processing without applying the searched augmentation policies and the Cutout. For a fair comparison, we report our resulting both using and without using the Enlarge Batch (EB) proposed by Adv. AA [41]. By leveraging $r_{EB} \times$EB in practice, the mini-batch size is $r_{EB}$ times larger, while the number of iterations is not changed. Besides, our searched policies have strong preferences, as only a few augmentation operations are preserved eventually, which is quite different from other methods like [4, 20, 41]. Details about them are presented in the supplementary material.

Table 1: **CIFAR-10 results.** Top-1 test error rates (%) are reported (lower is better). For fair comparison, we report our results both using and without using the Enlarge Batch proposed by Adv. AA [41]. We report Mean ± STD (standard deviation) of the test error rates wherever available.

| Approach | Res-18 | WRN | Shake-Shake | PyramidNet |
|---|---|---|---|---|
| Baseline | 4.66 | 3.87 | 2.86 | 2.67 |
| Cutout [7] | 3.62 | 3.08 | 2.56 | 2.31 |
| Fast AA [20] | - | 2.7 | 2.0 | 1.7 |
| RandAug [5] | - | 2.7 | 2.0 | 1.5 |
| AutoAug [4] | 3.46 | 2.68 | 1.99 | 1.48 |
| PBA [13] | - | 2.58 | 2.03 | 1.46 |
| OHL [21] | 3.29 | 2.61 | - | - |
| Adv. AA (8×EB) [41] | - | 1.90 ± 0.15 | 1.85 ± 0.12 | 1.36 ± 0.06 |
| Ours | 2.91 ± 0.062 | 1.95 ± 0.047 | 1.65 ± 0.039 | 1.31 ± 0.044 |
| Ours (8×EB) | **2.38 ± 0.041** | **1.57 ± 0.038** | **1.42 ± 0.040** | **1.24 ± 0.042** |

Table 2: **CIFAR-100 results.** Top-1 test error rates (%) are reported (lower is better). We report Mean ± STD (standard deviation) of the test error rates wherever available.

| Approach | WRN | Shake-Shake | PyramidNet |
|---|---|---|---|
| Baseline | 18.80 | 17.1 | 13.99 |
| Cutout [7] | 18.41 | 16.0 | 12.19 |
| Fast AA [20] | 17.3 | 14.6 | 11.7 |
| RandAug [5] | 16.7 | - | - |
| AutoAug [4] | 17.1 | 14.3 | 10.67 |
| PBA [13] | 16.7 | 15.3 | 10.94 |
| Adv. AA (8×EB) [41] | 15.49 ± 0.18 | 14.10 ± 0.15 | 10.42 ± 0.20 |
| Ours | 15.28 ± 0.067 | 14.07 ± 0.053 | **10.40 ± 0.040** |
| Ours (8×EB) | **14.16 ± 0.055** | **13.96 ± 0.032** | 10.45 ± 0.044 |

**Results on CIFAR**    The results on CIFAR-10 are summarized in Tab. 1. Comparing the results horizontally in Tab. 1, it can be seen that our learned policies using ResNet-18 could be well transferred to training other network models like WRN [39], Shake-shake [8] and PyramidNet [10]. Compared with the baseline without using the searched augmentation, the performance of all these models significantly improves after applying our searched policies for augmentation. Comparing the results vertically in Tab. 1, our AWS method is the best performer across all four network architectures.

Specifically, ours achieves the best top-1 test error of $1.24\%$ with PyramidNet+ShakeDrop, which is $0.12\%$ better than the second-best performer Adv. AA [41], even though we, unlike [41], do not use the Sample Pairing [14] for search. Consistent observations are found on the results on CIFAR-100 in Tab. 2. Ours again performs best on all four network architectures among the methods in comparison.

Table 3: **ImageNet results.** Top-1 / Top-5 test error rates (%) are reported (lower is better). We report Mean $\pm$ STD (standard deviation) of the test error rates of our method. "Ours" denotes our approach without using EB. "Ours ($4\times$ or $2\times$ EB)" denotes our approach using 4 times the mini-batch size for ResNet-50 and 2 times the mini-batch size for ResNet-200. Please note that Adv. AA used 8 times the mini-batch size.

| Approach | ResNet-50 | ResNet-200 |
|---|---|---|
| Baseline | 23.7 / 6.9 | 21.5 / 5.8 |
| Fast AA [20] | 22.4 / 6.3 | 19.4 / 4.7 |
| AutoAug [4] | 22.4 / 6.2 | 20.0 / 5.0 |
| RandAug [5] | 22.4 / 6.2 | - |
| OHL [21] | 21.07 / 5.68 | - |
| Adv. AA ($8\times$EB) | $20.60 \pm 0.15$ / $5.53 \pm 0.05$ | $18.68 \pm 0.18$ / $4.70 \pm 0.05$ |
| Ours | $20.61 \pm 0.17$ / $5.49 \pm 0.08$ | $18.64 \pm 0.16$ / $4.67 \pm 0.07$ |
| Ours ($4\times$ or $2\times$ EB) | $\mathbf{20.36 \pm 0.15}$ / $\mathbf{5.41 \pm 0.07}$ | $\mathbf{18.56 \pm 0.14}$ / $\mathbf{4.62 \pm 0.05}$ |

**Results on ImageNet**  The results on ImageNet are summarized in Tab. 3. We report both the top-1 and the top-5 test errors following the convention. Adv. AA [41] has evaluated its performance for different EB ratios $r \in \{2, 4, 8, 16, 32\}$. The test accuracy improves rapidly with the increase of $r$ up to 8. The further increase of $r$ does not bring a significant improvement. So $r = 8$ is finally used in Adv. AA. We tried to increase the batch size, but we can only use $4\times$EB for ResNet-50 and $2\times$EB for ResNet-200 due to the limited resources. As can be seen, we still achieve superior performance over those automatic augmentation works in comparison. Moreover, our outstanding performance using the heavy model ResNet-200 also verifies the generalization of our learned augmentation policies.

Table 4: **Comparison with the computation cost.** We estimate ours with Tesla V100.

| Approach | Cutout [7] | AutoAug [4] | OHL [21] | Adv. AA [41] | Ours |
|---|---|---|---|---|---|
| Time Consuming (times) | - | 60 | 1 | 5 | 1.5 |
| Relative Error Reduction (%) | 0 | 12.99 | 15.26 | 38.31 | 49.02 |

**Results on computational cost**  We further compare the computational cost among different auto-augmentation methods and report the error reductions of WRN relative to Cutout's. Following the existing works, the computation costs on CIFAR-10 are reported in Tab. 4. In this table, we use the GPU hours used by OHL-Auto-Aug [21] as the baseline, and report the relative time consuming on this baseline. As can be seen that our method is 40 times faster than AutoAugment [4] with an even better performance. Although it is slightly slower than OHL, our method has a salient performance advantage over it. Overall, our proposed method is a very promising approach: it has the best performance with acceptable computational cost.

### 4.4 Comparison Among Proxy Tasks

To verify the superiority of the proxy selected by us, we make comparisons among different proxy tasks using ResNet-18 on CIFAR-10. To solve the problem in Equ. 4, we design four optional proxies, which are summarized in Tab. 5. The correlations between $\text{ACC}(\bar{\omega}_\theta^*)$ and $\text{ACC}(\omega^*)$ are investigated, shown in Fig. 4. As can be seen in Tab. 5, among the four options, our selection ($P_{AF}$) produces the highest Pearson correlation coefficient $0.85$, which outperforms other options by a large margin. Specifically, the proxy $P_{NF}$ trains the model without data augmentation in the first stage, and only searches the augmentation policy in the second stage like AutoAugment [4]. Its inferior performance to our proposed proxy $P_{AF}$ may suggest that simply performing AutoAugment [4] only in the late stage could not lead to good results. As for the proxy $P_{IT}$, there is no first-stage training and the

network parameters in the second stage are randomly initialized. Its inferior performance to $P_{AF}$ may suggest that less trained network parameters could not generate a reliable ranking for rewording. Finally, the proxy $P_{AV}$ is similar to that used in Fast AutoAugment [20] and the low correlation coefficient indicates that the evaluation process without fine-tuning may be unreliable.

Table 5: **Optional proxy tasks**. The first three proxies fine-tune ($P_{IT}$ does not learn $\omega_{share}$) $\omega_{share}$ in different ways with the same goal to maximize the validation accuracy. The last proxy aims to maximize the accuracy on an *augmented* validation set without fine-tuning.

| Symbolic representation | The way to obtain $\omega_{share}$ | The way to optimize | pearsonr |
|---|---|---|---|
| $P_{AF}$ (ours) | Train with Augmentation | Finetune+Eval | 0.85 |
| $P_{NF}$ | Train without Augmentation | Finetune+Eval | 0.55 |
| $P_{IT}$ | Random Initialized | Finetune+Eval | 0.36 |
| $P_{AV}$ | Train with Augmentation | Eval | 0.045 |

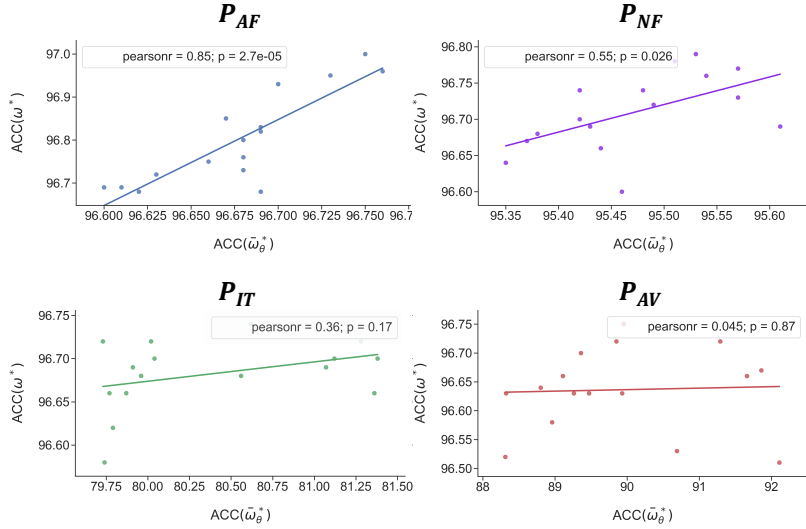

Figure 4: **Correlation between ACC($\bar{\omega}_\theta^*$) and ACC($\omega^*$)**, where $\omega^*$ denotes the optimal network parameters for a fixed augmentation policy $\theta$ and $\bar{\omega}_\theta^*$ denotes the network parameters obtained by a proxy task via finetuing the checkpoint. Four proxy tasks, $P_{AF}$, $P_{NF}$, $P_{IT}$ and $P_{AV}$ are investigated.

## 5 Conclusion

In this paper, we propose an innovative and elegant way to search for auto-augmentation policies effectively and efficiently. We first verify that data augmentation operations function more in the late training stages. Based on this phenomenon, we propose an efficient and reliable proxy task for fast evaluation of augmentation policy and solve the auto-augmentation search problem with an augmentation-wise weight sharing proxy. The intensive empirical evaluations show that the proposed AWS auto-augmentation outperforms both previous searched and handcrafted augmentation policies. To our knowledge, it is the first time for a weight sharing proxy paradigm to be applied to augmentation search. The augmentation policies we found on both CIFAR and ImageNet benchmark will be released to the public as off-the-shelf augmentation policies to push the boundary of the stat-of-the-art performance.

## Broader Impact

In this paper, we propose a new framework to conduct an efficient and reliable Automated Augmentation (AutoAug) search and achieve superior performance compared with existing methods. AutoAug enhances the performances of deep models as a typical Automated Machine Learning (AutoML) technique.

For fundamental research and ML applications, our research contributes towards many computer vision areas that benefit from image data augmentations. It may help reduce the demand for data scientists by enabling domain experts to automatically design tailored augmentation strategies without extensive knowledge of statistics and machine learning.

For broader societal implications, as an AutoML technique, our approach can be utilized to build models and establish reasonable lower bounds of them for performance quickly and cheaply. It may be useful and powerful to ML practitioners in various entities, such as the media industry, the transportation industry, and the automatic production industries. However, each of these uses may result in job losses. Some other issues, like personal privacy leak problems, may also be raised when this technique is used by those malicious. In summary, this technique may be socially beneficial or harmful, which depends on the users. We would encourage the researchers, general practitioners, or anyone else to use it for social benefits, rather than infringe the interests of individuals and the nation, and threaten social stability.

## Acknowledgments and Disclosure of Funding

Wanli Ouyang is supoorted by the Australian Research Council Grant DP200103223, and Australian Medical Research Future Fund MRFAI000085. Support from them as well as numerous colleagues, friends and experts from SenseTime and University of Sydney is gratefully acknowledged. We also appreciate all the reviewers for their valuable and constructive suggestions.

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
