[Supplementary Material]

# A    Ablation Study

In this ablation study, we further investigate the power of the policies searched by our approach and the closely related method AutoAug [1]. We rank the augmentation operations based on their probabilities in decreasing order. Therefore, the operations ranked on the top could be deemed as the most important augmentations. Then we gradually remove the most important operations from the searched policy one by one and investigate the change of the Top-1 test error rates, as reported in Tab. 1. As can be seen, when the most important operations are removed gradually, the performance of the AutoAug remains similar. On the contrary, during this process, the performance of ours drops significantly. This shows that the augmentations in our policy are much more powerful than those in the AutoAug.

Table 1: **Ablation Study.** Top-1 test error rates (%) is reported (lower is better). We report Mean $\pm$ STD (standard deviation) of the test error rates.

| Approach | Apply All | Without Top 1 | Without Top $1 \sim 2$ | Without Top $1 \sim 3$ |
|---|---|---|---|---|
| AutoAug [1] (our impl.) | $3.40 \pm 0.070$ | $3.45 \pm 0.066$ | $3.36 \pm 0.065$ | $3.46 \pm 0.082$ |
| Ours | $2.91 \pm 0.062$ | $3.10 \pm 0.056$ | $3.13 \pm 0.099$ | $3.19 \pm 0.110$ |

## B Proof

The proof of the proposition in Sec. 3.3 is as follows:

**Proof.** The KL-divergence between $p_{\theta\theta}$ and $p_{\bar{\theta}\theta}$ is as follows:

$$
\begin{aligned}
D_{\mathcal{KL}}(p_{\theta\theta} \parallel p_{\bar{\theta}\theta}) &= \Sigma_\tau(p_{\theta\theta}(\tau) \log \frac{p_{\theta\theta}(\tau)}{p_{\bar{\theta}\theta}(\tau)}) \\
&= \Sigma_\tau(p_{\theta\theta}(\tau) \log \frac{\Pi_{i=1}^N p_\theta(\tau_i)}{\Pi_{i=1}^K p_{\bar{\theta}}(\tau_i) \, \Pi_{i=K+1}^N p_\theta(\tau_i)}) \\
&= \Sigma_\tau(p_{\theta\theta}(\tau) \log \frac{\Pi_{i=1}^K p_\theta(\tau_i)}{\Pi_{i=1}^K p_{\bar{\theta}}(\tau_i)}) \\
&= \Sigma_\tau(p_{\theta\theta}(\tau)\Sigma_{i=1}^K \log p_\theta(\tau_i)) \; - \; \Sigma_\tau(p_{\theta\theta}(\tau)\Sigma_{i=1}^K \log p_{\bar{\theta}}(\tau_i)) \,.
\end{aligned}
\tag{1}
$$

Since $\Sigma_\tau(p_{\theta\theta}(\tau)\Sigma_{i=1}^K \log p_\theta(\tau_i))$ is constant with respect to $\bar{\theta}$, the $\bar{\theta}^*$ that minimizes $D_{\mathcal{KL}}(p_{\theta\theta} \parallel p_{\bar{\theta}\theta})$ should satisfy:

$$
\bar{\theta}^* = \arg\max_{\bar{\theta}} \; \Sigma_\tau(p_{\theta\theta}(\tau)\Sigma_{i=1}^K \log p_{\bar{\theta}}(\tau_i)) = \arg\max_{\bar{\theta}} \; \Sigma_\tau(p_{\theta\theta}(\tau)\frac{1}{K}\Sigma_{i=1}^K \log p_{\bar{\theta}}(\tau_i)) \,.
\tag{2}
$$

By Jensen's inequality with the strictly concave function $\log(\cdot)$, we have:

$$
\Sigma_\tau(p_{\theta\theta}(\tau)\frac{1}{K}\Sigma_{i=1}^K \log p_{\bar{\theta}}(\tau_i)) \leq \Sigma_\tau(p_{\theta\theta}(\tau) \log(\Sigma_{i=1}^K \frac{1}{K}p_{\bar{\theta}}(\tau_i))) \,.
\tag{3}
$$

The equality holds if and only if $p_{\bar{\theta}}(\tau_1) = p_{\bar{\theta}}(\tau_2) = \cdots = p_{\bar{\theta}}(\tau_K)$ for all the possible $\tau$. In other words, the KL divergence is minimized when $\bar{\theta}$ is uniform sampling. $\qquad\square$

## C Investigation of the Number of Epochs of Fine-tuning

Figure 1: We investigate the key hyper-parameter $N_{late}$ by visualizing the difference it brings to the search dynamics. All the experiments are conducted with ResNet-18 [4] on CIFAR-10 [6]. The Pearson correlation coefficient ($r$) between $ACC(\bar{\omega}_\theta^*)$ and $ACC(\omega^*)$ are annotated in the figure.

We investigate different numbers of epochs in the late training stage($N_{late}$). By adjusting $N_{late}$ we can still maintain the reliability of policy evaluation to a large extent. The results are shown in Fig. 1. We find that the policy optimization becomes hard to converge when a small $N_{late}$ is used, as the performances among different $\bar{\omega}_\theta^*$ are too close. And when a large $N_{late}$ is used, $ACC(\bar{\omega}_\theta^*)$ is notably higher. However, the final performance does not benefit from this, as the correlation between $ACC(\bar{\omega}_\theta^*)$ and $ACC(\omega^*)$ does not change much between $N_{late} = 10$ and $N_{late} = 30$. Thus we choose $N_{late} = 10$ as the final configuration for the efficiency.

# D Augmentation Elements

The augmentation elements are listed as follows. We use almost the same elements as AutoAug's [1]. But we do not introduce Cutout [3] and Sample Pairing [5] into the search space.

Table 2: **List of Candidate Augmentation Elements.**

| Elements | Ranges of Magnitude |
|---|---|
| Horizontal Shear | $\{0.1, 0.2, 0.3\}$ |
| Vertical Shear | $\{0.1, 0.2, 0.3\}$ |
| Horizontal Translate | $\{0.15, 0.3, 0.45\}$ |
| Vertical Translate | $\{0.15, 0.3, 0.45\}$ |
| Rotate | $\{10, 20, 30\}$ |
| Color Adjust | $\{0.3, 0.6, 0.9\}$ |
| Posterize | $\{4.4, 5.6, 6.8\}$ |
| Solarize | $\{26, 102, 179\}$ |
| Contrast | $\{1.3, 1.6, 1.9\}$ |
| Sharpness | $\{1.3, 1.6, 1.9\}$ |
| Brightness | $\{1.3, 1.6, 1.9\}$ |
| Autocontrast | None |
| Equalize | None |
| Invert | None |

 # E  Datasets Splitting Details

Table 3: **Datasets Splitting Details.** On both of the two CIFAR [6] datasets, we use a validation set of 10,000 images, which is randomly split from the original training set, which contains 50,000 images, to calculate the validation accuracy during the searching. For ImageNet [2], we use a reduced subset of ImageNet train set when searching the policies, although our method is affordable to be directly performed on ImageNet. This subset contains $128,000$ images and $500$ classes (randomly chosen). We also set aside a validation set (no intersection with the reduced train subset and containing the same $500$ classes) of 50,000 images split from the training dataset for getting the validation accuracy.

| Dataset | Train Set Size | Validation Set Size | Test Set Size |
|---|---|---|---|
| CIFAR-10 [6] | 40,000 | 10,000 | 10,000 |
| CIFAR-100 [6] | 40,000 | 10,000 | 10,000 |
| Reduced ImageNet [2] | 128,000 | 50,000 | 50,000 |

# F More Implementation Details

**CIFAR**    Once the policies have been learned, they are applied to training the models again from scratch, as well as another network models for the investigation of the transferability of the policies between different network models. For ResNet-18 and Wide-ResNet-28-10, we use a mini-batch size of 256 and the SGD with a Nesterov momentum of 0.9. The weight decay is set to $0.0001$, and the cosine learning rate scheme is utilized with the maximum learning rate of $0.4$. The number of epochs is set to 300. For PyramidNet+ShakeDrop and Shake-Shake ($26\ 2 \times 32$d), we use the same settings as those in [7].

For a fair comparison among different augmentation methods, we apply a basic pre-processing following the convention of the state-of-the-art CIFAR-10 models: standardizing the data, random horizontal flips with 50% probability, zero-padding and random crops, and finally Cutout [3] with $16 \times 16$ pixels. During our comparison, the searched policy is applied on top of this basic pre-processing step. That is, for each input training image, the basic pre-processing is first performed, then the policies learned by an augmentation method, and finally the Cutout.

**ImageNet**    Once the policies have been obtained, they are applied to training ResNet-50 from scratch, as well as another network model ResNet-200 for the study of policy transferability. The hyper-parameters used to train ResNet-50 and ResNet-200 are the same as those in [1] except a cosine learning rate scheduler. Moreover, our learned policies are applied on top of a standard Inception-style pre-processing, which includes standardizing, random horizontal flips with 50% probability, and random distortions of colors [8]. This pre-processing step is uniformly applied to all the methods in comparison.

## G    Details of Searched Policies.

Figure 2: **The changes of probability distributions of the searched policies on CIFAR-10 (the left) and ImageNet (the right) over time.**

We visualize the changes of probability distributions of the searched policies on CIFAR-10 and ImageNet over time. We calculate the marginal distribution parameters of the first element in our augmentation operations. As shown in the picture, our searched policies have strong preferences, as only a few augmentation operations are preserved eventually, and probabilities of many other operations are close to zero, which is quite different from other methods like [1, 7, 9].