[Reviews · NeurIPS 2020]

Review 1

Summary and Contributions: POST REBUTTAL: After reading the authors' response and discussing with other reviewers, I decided to keep my score. In this paper, the authors propose Augmentation-Wise Weight Sharing (AWS), a weight sharing strategy to efficiently search for data augmentation operations in image classification. AWS is based on a simple observation: data augmentation is more effective later in the training process, rather than earlier. Thus, AWS trains a single model for a while, and then only performs data augmentation search for a few last epochs, all starting from the trained shared weights. I think this is such a simple and elegant observation. The authors report strong empirical results on CIFAR-10, okay-ish generalization to CIFAR-100, and miss-report the results for ImageNet. Please see my comments in Weaknesses for more details. Overall, I think the paper contributes something nice to the field. Spotting similar patterns, ie. where and when does an AutoML process matter the most, and sharing the parts that don’t matter, is a very simple and elegant takeaway. However, the authors do need to include a more fair comparison against existing search approaches and update their claims accordingly. If this is not done, I would be very uncomfortable seeing this paper being accepted, even though I really like the method and trust that it works.

Strengths: [Elegant, well-motivated, and well-presented method] The whole AWS scheme is very elegant. The empirical gains on CIFAR-10 are impressive. While the gains do not carry over to CIFAR-100 and to ImageNet, I believe that the authors simply don’t have the resources to tune extensively on these datasets.

Weaknesses: [Missing empirical results from previous work] In Table 2, why did the authors omit AdvAA, which they present in Table 1? I checked the AdvAA paper, and they obtained 20.6% test error on ImageNet with ResNet-50, ie. the same performance with AWS. The authors should add this comparison. Sure, it makes the results of AWS look weaker, but it wouldn’t devalue AWS.

Correctness: I could trust the correctness of the paper’s results.

Clarity: The paper is clearly written and the method is easy to understand.

Relation to Prior Work: I think the paper appropriately cites related works.

Reproducibility: Yes

Additional Feedback:


Review 2

Summary and Contributions: The authors propose to improve AutoAugment, a prior work to acquire augmentation automatically, with the Augmentation-Wise Weight Sharing scheme.

Strengths: 1. Good performance. 2. The finding in Figure 2 is good. The motivation for the Augmentation-Wise Weight Sharing scheme is clear. 3. The method is easy to follow and makes sense.

Weaknesses: 1. In Fig 1, it is used to validate the claim "Specifically, the compromised evaluation process would distort the ranking for augmentation strategies since the model trained with too few iterations are unstable." However, the Figure cannot support the claim: the change of rank cannot indicate that the model is unstable. It just shows that the improvements caused the three methods are different. The rank decreases but the accuracy may also get improved. Also, whether a model is stable or not needs more clarification. 2. Overall, the proposed method is an incremental improvement on the existing AutoAugment. But the performance improvements are not very significant when comparing to recent methods. On ImageNet, resnet-50 is only 0.3 better than [18]. Also, the std is very large, which leads to doubts about generalizability. 3. In table 1, it is said that "We report Mean STD (standard deviation) of the test error rates wherever available." But no std reported in the table.

Correctness: correct

Clarity: Well written

Relation to Prior Work: Clearly discussed

Reproducibility: Yes

Additional Feedback: I've read other reviewers' reviews and the rebuttal. Most of my concerns have been addressed. I increased my score.


Review 3

Summary and Contributions: This work observes that the augmentation is better to be applied to the late stage rather than the early stage. Based on this observation, it proposes to share early stage weights and finetune it in the augmentation searching process. This design significantly speeds up the searching and optimization, and also reaches SOTA performance.

Strengths: The key observation in this paper is interesting, and is empirically evaluated. The weight sharing idea is not new, but is naturally combined with the observation in the proposed pipeline. On CIFAR it achieves significant improvement. The supplementary materials include detailed information on experiment setup.

Weaknesses: - The ResNet50 accuracy on ImageNet looks marginal: 20.73+0.17 = 20.9, which is close to 21.07 from OHL. Would be better to compare with the Enlarge Batch trick. - This paper makes the observation based on image classification task, which is training from scratch. It would also be important to evaluate that if transfer learning with a pre-trained backbone still have this phenomenon. Update after Rebuttal: I read the opinions from other reviewers, and the feedback from the authors. The feedback is valid, but not strong enough to improve my rating, thus I'd like to keep my score.

Correctness: The proof, claims, and the empirical settings are correct to me.

Clarity: The paper is easy to follow.

Relation to Prior Work: Yes

Reproducibility: Yes

Additional Feedback: - It is mentioned to be unaffordable to use the enlarge batch trick, and it is fair. Still, there's a workaround to approximate large batch size training: in training let the model performs forward and backward for several batches and accumulates the gradient, then only apply the gradient to weights once the accumulated number of batches is at least 16,384. - Evaluate the key observation for other tasks involving transfer learning.

[Author Response · NeurIPS 2020]

We sincerely thank all the reviewers for their careful readings and valuable comments. We will address the reviewers'
comments point by point.

Table A: **Updated part of ImageNet results.** In the paper for our original experiment on ImageNet, we use the same
hyper-parameters of that for CIFAR10. In the new experiment we only change the learning rate of RL for agent from
0.1 to 0.2. The resulting policy is used to train Res50 and Res200 for Ours (without EB) and Ours ($4\times$ or $2\times$ EB). By
leveraging EB with a batch size ratio $r$ ($r\times$EB), the mini-batch size is $r$ times larger while the number of iterations is
not changed (line 240). *Ours (without EB)* denotes our approach without using EB. *Ours ($4\times$ or $2\times$ EB)* denotes our
approach using 4 times the mini-batch size for Res50 and 2 times the mini-batch size for Res200. Please note that *Adv.
Aug* used 8 times the mini-batch size.

| Approach | Res50 | Res200 |
|---|---|---|
| OHL (without EB) | 21.07 / 5.68 | - |
| Adv. Aug ($8\times$EB) | $20.60 \pm 0.15$ / $5.53 \pm 0.05$ | $18.68 \pm 0.18$ / $4.70 \pm 0.05$ |
| Ours (without EB) | $20.61 \pm 0.17$ / $5.49 \pm 0.08$ | $18.64 \pm 0.16$ / $4.67 \pm 0.07$ |
| Ours ($4\times$ or $2\times$ EB) | $\mathbf{20.36 \pm 0.15}$ / $\mathbf{5.41 \pm 0.07}$ | $\mathbf{18.56 \pm 0.14}$ / $\mathbf{4.62 \pm 0.05}$ |

**R#1 Q1**: In Table 2, why did the authors omit AdvAA? The authors should add this comparison.

**A: 1)** In the caption of Table 2, we explained that AdvAA is not included here since the EB used in AdvAA is not used
in the other compared approaches in Table 2, making it unfair to compare AdvAA with other approaches and ours.
**2)** AdvAA has evaluated its performance for different EB ratios $r \in \{2, 4, 8, 16, 32\}$. The test accuracy improves
rapidly with the increase of $r$ up to 8. The further increase of $r$ does not bring a significant improvement. So $r = 8$ is
finally used in AdvAA. Following **R#4**'s advice on the mini-batch size, we tried to increase the batch size, but we can
only use $4\times$EB for Res50 and $2\times$EB for Res200 due to the limited resources. As shown in Table. A, our approach can
still achieve performance on a par with AdvAA without applying EB. And we've achieved better performance with
even smaller $r$ on both Res50 and Res200 backbones. We will update the results and some claims accordingly in our
revision. Thanks for the kind suggestion.

**R#3 Q1**: In figure 1, the change of rank cannot indicate that the model is unstable ...

**A**: Sorry for the confusion. We will change into: ... the compromised evaluation process would distort the ranking for
augmentation strategies since the rank for the models trained with too few iterations are known to be inconsistent with
the final models trained with sufficient iterations.

**R#3 Q2**: The performance improvements are not very significant when comparing to recent methods. On ImageNet,
resnet-50 is only 0.3 better than OHL. Also, the std is very large.

**A: 1)** On CIFAR10, the performance gains of our AWS are substantial. On CIFAR100 and ImageNet, AWS still achieves
superior performance over many recent works without tuned hyper-parameters, i.e., using the same hyper-parameters as
CIFAR10's. **2)** On ImageNet, OHL stated in the paper that they manually tuned the agent learning rate and used the
best one but we did not. We conducted a new search by simply doubling the agent learning rate of ours. Table. A shows
AWS achieves around 0.5% accuracy improvement over OHL. **3)** Besides, we would like to point out that, rather than
being 'very large', our std is comparable with AdvAA's, while many other works do not report their std.

**R#3 Q3**: In table 1 ... no std reported.

**A**: Sorry for the confusion. In line 229 we stated that our test error rates with std are given in the supplementary (line
52), which conflicted with the caption of table 1. We will revise the caption of Table 1.

**R#4 Q1**: 20.73+0.17 = 20.9, which is close to 21.07 from OHL.

**A: 1)** This comparison would be unfair. Please note that, 20.73+0.17 is the worst
case for our AWS, while 21.07 could be the mean or even best performance for
OHL since it did not report std. **2)** The updated result in Table. A shows that
AWS achieves around 0.5% accuracy improvement over OHL. Please refer to
**R#3 Q2 2)** for more details.

**R#4 Q2**: Evaluate the key observation for other tasks ...

**A**: Thanks for the valuable advice. We've evaluated our key observation in the
'pre-train fine-tune' paradigm. We first pre-train the Res18 on CIFAR100 for 200
epochs without augmentation. Then we transfer it to CIFAR10 to train (fine-tune)
100 epochs and conduct a similar experiment in Sec. 3.1.1. Figure. A is similar
to Fig. 2 in our paper, showing that data augmentation is more effective in the
later training process. Furthermore, we agree that further research can be conduct
based on this work. As our future work, we will evaluate our key observation for
more different tasks, such as object detection and semantic segmentation.

Figure A: The investigation on a transfer learning task.

[Meta-Review · NeurIPS 2020]

The simple and elegant insight into framing data augmentation as a bilevel optimization problem as is often done in NAS and observing that later stages result in the most consistent gains and stable rankings thus allowing the design of a fast but accurate proxy task for the inner loop of bilevel opt. was well appreciated. Thorough experiments and clear writing also makes it stand out. Authors are encouraged to take all reviewer feedback to further improve the paper for publication.